# Use of Smartphones for the Detection of Uterine Cervical Cancer: A Systematic Review

**DOI:** 10.3390/cancers13236047

**Published:** 2021-12-01

**Authors:** Denisse Champin, Max Carlos Ramírez-Soto, Javier Vargas-Herrera

**Affiliations:** 1Facultad de Ciencias de la Salud, Universidad Tecnológica del Peru, Lima 15046, Peru; dchampin@utp.edu.pe; 2Departamento de Telemedicina, Facultad de Medicina, Universidad Nacional Mayor de San Marcos, Lima 15001, Peru; jvargash@unmsm.edu.pe

**Keywords:** cervical cancer, smartphone, colposcopy, visual inspection with acetic acid, visual inspection with Lugol’s iodine

## Abstract

**Simple Summary:**

Several studies reveal that digital images taken with a smartphone after a visual inspection with acetic acid (VIA) or Lugol’s iodine (VILI) may be useful for detecting cervical intraepithelial neoplasia. Therefore, smartphones could be useful in the early detection of uterine cervical lesions and an alternative to colposcopy in countries with limited health resources. In this systematic review, we found that the VIA using a smartphone seems to be more sensitive than the VIA, VILI, or VIA/VILI examinations with the naked eye. Therefore, it can improve diagnostic accuracy for the detection of uterine cervical lesions.

**Abstract:**

Little is known regarding the usefulness of the smartphone in the detection of uterine cervical lesions or uterine cervical cancer. Therefore, we evaluated the usefulness of the smartphone in the detection of uterine cervical lesions and measured its diagnostic accuracy by comparing its findings with histological findings. We conducted a systematic review to identify studies on the usefulness of the smartphone in detecting uterine cervical lesions indexed in SCOPUS, MEDLINE/PubMed, Cochrane, OVID, Web of Science, and SciELO until November 2020. The risk of bias and applicability was assessed using the Quality Assessment of Diagnostic Accuracy Studies-2 tool. A total of 16 studies that evaluated the usefulness of the smartphone in the detection of uterine cervical lesions based on the images clicked after visual inspection with acetic acid (VIA), Lugol’s iodine (VILI), or VIA/VILI combination were included in the study. Five studies estimated diagnostic sensitivity and specificity, nine described diagnostic concordance, and five described the usefulness of mobile technology. Among the five first studies, the sensitivity ranged between 66.7% (95% confidence interval (CI); 30.0–90.3%) and 94.1% (95% CI; 81.6–98.3%), and the specificity ranged between 24.0% (95% CI; 9.0–45.0%) and 85.7% (95% CI; 76.7–91.6%). The risk of bias was low (20%), and the applicability was high. In conclusion, the smartphone images clicked after a VIA were found to be more sensitive than those following the VILI method or the VIA/VILI combination and naked-eye techniques in detecting uterine cervical lesions. Thus, a smartphone may be useful in the detection of uterine cervical lesions; however, its sensitivity and specificity are still limited.

## 1. Introduction

Despite advances in the diagnosis of uterine cervical cancer (UCC) and the administration of vaccines against the high-risk human papillomavirus (HPV-16 and 18), UCC remains one of the leading causes of death [1]. UCC can be prevented by the early detection of cervical intraepithelial neoplasia (CIN) based on a PAP test or the detection of high-risk HPV through a PCR test. However, due to economic barriers and lack of availability of resources for these tests, the WHO recommends using visual inspection with acetic acid (VIA) [2].

Colposcopy is the standard technique used for determining the degree of uterine cervical lesions and high-grade CIN or abnormal cytology. The performance of this procedure varies based on the level of care, experience of the colposcopist, and the type of colposcopy (digital, optical, or video colposcopy). The disadvantage of this procedure is that the colposcope and colposcopists are not always available at primary care facilities.

Recently, several studies revealed that digital images taken with a smartphone after a VIA or Lugol’s iodine (VILI) may be useful for detecting CIN2 [3,4]. Another study found a correlation between smartphone-based histological diagnoses and colposcopic findings in women with abnormal cervical cytology [5]. According to these reports, the main advantage of smartphones is that they are easy to use and a low cost way to capture digital images that can be sent to an expert colposcopist for review in real time compared with conventional colposcopy, which is expensive and not always available at all levels of care. Therefore, smartphones could be useful in the early detection of uterine cervical lesions and an alternative to colposcopy in countries with limited health resources.

Despite the fact that the use of smartphones and apps in public health is almost universal, studies evaluating the usefulness of smartphones in the detection of UCC are still limited. This suggests there is a need for reviewing and synthesizing the published literature with a view to creating consensus and providing recommendations for the use of smartphones in the field. Therefore, the objective of this study was to evaluate the usefulness of the smartphone in the detection of uterine cervical lesions and measure its diagnostic accuracy by comparing its findings against histological findings.

## 2. Materials and Methods

A systematic review was performed according to the Preferred Reporting Items for a Systematic Review and Meta-analysis (PRISMA) guidelines (Appendix A, Table A1) [6]. The study protocol was not registered in PROSPERO.

### 2.1. Literature Research

The SCOPUS, MEDLINE/PubMed, Cochrane, OVID, Web of Science, and SciELO databases were searched for articles published between 1 January 2010 and 30 September 2020. There were no language restrictions. The following keywords were used: “Smartphone,” “cervical cancer,” “uterine cervical cancer,” “mobile,” “mobile technology,” “tele-cytology,” “technology and prevention,” “diagnosis,” “sensibility,” and “specificity.” Boolean operators (AND, OR, and NOT) and truncators (“”, (), and *) were used to combine the keywords (Table 1). In addition, the references of the selected studies were reviewed.

### 2.2. Definition

Diagnostic tests using mobile technology are defined as diagnostic procedures or diagnostic tests that include the use of mobile devices to capture images after performing a diagnostic procedure; for example, images taken with a smartphone after the application of acetic acid or Lugol’s iodine to detect uterine cervical lesions.

### 2.3. Inclusion and Exclusion Criteria

We included cohort, case-control, randomized trials, and pilot and field studies that evaluated the usefulness of the smartphone in the detection of uterine cervical lesions and/or measured its diagnostic precision (sensitivity and specificity). Review articles, case reports, editorials, animal studies, and studies that only reported on the development of health technologies for the diagnosis of UCC were excluded.

Three investigators (D.C., M.C.R.-S., and J.V.-H.) screened the titles and abstracts of the articles identified in the searches independently. The full text of the articles of interest was reviewed by the three investigators. Articles that met the inclusion criteria were included in the study. Disagreements about an article were resolved by consensus at the full-text review stage. The following information was extracted from each selected study: author, year, country, design, study population, the diagnostic methods used (PAP, VIA, or VILI, histological analysis, and colposcopy), mobile technology used, intervention and control group, and the diagnostic accuracy of the test results of the index and reference test (sensitivity, specificity).

### 2.4. Quality of Evidence Analysis

Three reviewers (D.C., M.C.R.-S., and J.V.-H.) independently assessed the risk of bias and applicability using the Quality Assessment of Diagnostic Accuracy Studies (QUADAS-2) tool [7]. The patient selection, index test performance, reference test performance, and flow and time (only for the risk of bias) domains were used. Quality assessment was performed for each study that evaluated the usefulness of the smartphone in the detection of uterine cervical lesions and measured the diagnostic sensitivity and specificity. Disagreements between reviewers were resolved by consensus.

### 2.5. Evidence Synthesis

We synthesized the evidence descriptively. The sensitivity and specificity values of the smartphone-based diagnostic tests vs. reference tests (histology) were compared. As a secondary outcome, a descriptive synthesis of the cross-sectional studies that evaluated the usefulness of the smartphone for detecting uterine cervical lesions without a comparison group or one that evaluated the concordance between the observers was performed. This study was presented to and approved by the Ethics and Research Committee of the Universidad Tecnológica del Peru.

### 2.6. Role of the Funding Source

The funder of the study had no role in study design, data collection, data analysis, data interpretation, or writing of the report. All authors had full access to all the data in the study.

## 3. Results

Figure 1 shows the selection of studies. A total of 707 studies were retrieved during the systematic review. After excluding duplicate studies and studies that did not meet the inclusion criteria, 16 studies were selected, including four retrieved from the reference review of the selected articles [3,4,5,8,9,10,11,12,13,14,15,16,17,18,19,20].

### 3.1. Evidence Synthesis

Five of the studies estimated diagnostic sensitivity and specificity. In these studies, the risk of bias and applicability were assessed. The five studies were assessed using QUADAS-2, and the results are summarized in Figure 2A,B. A low risk of bias (20%) and 100% applicability was observed for the domains of patient selection, reference standard, and the index test (Figure 2A,B).

Four studies were from Madagascar, and one was from Japan. The participants in the four studies had ≥2 CIN lesions. The degree of heterogeneity was high for the sensitivity and specificity results. The five studies’ sensitivity ranged between 66.7% (95% confidence interval (CI); 30.0–90.3) and 94.1% (95% CI; 81.6–98.3), and the specificity ranged between 24.0% (95% CI; 9.0–45.0) and 85.7% (95% CI; 76.7–91.6) (Figure 3).

The sensitivity of the VIA/VILI combination in three studies ranged between 66.7% (95% CI; 30.0–90.3%) and 71.4% (95% CI; 29.0–96.0%), and the specificity ranged between 62.4% (95% CI; 57.5–67.4%) and 85.7% (95% CI; 76.7–91.6%). The VIA sensitivity in two studies ranged between 92.0% (95% CI; 81.0–98.0%) and 94.1% (95% CI; 81.6–98.3%), and the specificity ranged between 24.0% (95% CI; 9.0–45.0%) and 50.4% (95% CI; 35.9–64.8%). Finally, the VILI test’s sensitivity and specificity rates were 78.8% (95% CI; 54.1–92.1%) and 56.4% (95% CI; 38.2–72.9%), respectively (Figure 3).

### 3.2. Descriptive Analyses

Nine studies determined diagnostic agreement or agreement between healthcare personnel capturing and interpreting images with smartphones (Table 2). For example: Catarino et al. [4] and Ricard-Gauthier et al. [8] evaluated the agreement between on-site physicians and the consensus of off-site physicians. Tanaka et al. [5] evaluated the correlation between the histological diagnosis based on samples obtained using a smartphone versus that based on colposcopy samples. Two studies compared agreement between nurses and physicians in reading smartphone images after a VIA (45%, Sharma et al.) [9] and (82%, Quinley et al.) [10]. One study evaluated concordance between the images captured with the mobile colposcope (smartcopy) vs. the standard colposcope [11]. Another study found a high degree of concordance (79.5%) when comparing the Swedish scoring results between the mobile (gynocular) colposcope vs. the standard colposcope (Singh et al.) [12]. On the other hand, another study evaluated the quality of the images taken with a smartphone for the detection of uterine cervical lesions among three gynecologists who were colposcopy experts (45%, Gallay et al.) [13]. Finally, in one study, we found a high degree of concordance between microscopic images versus images taken with a smartphone [14].

Table 3 summarizes the results of five studies in which the usefulness of mobile technology for the detection of cervical lesions is evaluated. Their origins are varied, including Japan, USA, and African cities. The mobile colposcope was evaluated in two of these studies [15,16]. The first evaluated a system to improve the monitoring and follow-up of screening with the introduction of a clinical decision tree in software that was incorporated into the mobile colposcope. The other evaluated the use of an improved mobile colposcope operated by expert colposcopists. Two studies by one author [17,18] in Tanzania show that training health personnel markedly improved diagnostic acuity. In the most recent publication, an enhanced VIA platform for smartphones was incorporated to secure the real-time exchange of cervical images for remote support supervision and data monitoring and evaluation [20].

## 4. Discussion

Subsection: The review of the 16 publications allowed us to identify five studies that had similar characteristics. These studies used a basic smartphone as a technological tool to obtain images of the cervix, following which they performed the VIA and/or VILI examination [3,4,5,8,9,10,11,12,13,14,15,16,17,18,19,20]. The histology was standard, and the on-site and off-site operators were physicians that reported sensitivity and specificity [3,4,5,8,19].

In our study, the sensitivity for the detection of ≥2 CIN lesions with VIA/VILI using a smartphone was found to be low (66.7% and 71.4%) compared with a sensitivity rate of 75% (95% CI; 69–81) for VIA/VILI with naked-eye observation determined in a meta-analysis by Catarino et al. [21]. This difference can probably be explained by the average estimate of the high sensitivity of VIA versus a low VILI sensitivity, which might result in a decrease in diagnostic sensitivity. Unlike that reported in previous meta-analyses by Qiao et al. (73.2%) [22], Catarino et al. (78%) [21], and Arbyn et al. (79%) [23], where they used VIA with the naked eye, in our systematic review, VIA using a smartphone was found to be more sensitive at detecting uterine cervical lesions (>90%) [3,5]. However, although the sensitivity of VILI with the naked eye is high, in this systematic review, we found that the sensitivity of VILI using a smartphone was lower (78.8%). We also found that VIA was more sensitive than VILI, in contrast to two meta-analyses, wherein the sensitivity of VILI was higher than VIA [19,20]. Similarly, we found that VIA was more sensitive than the VIA/VILI combination, which another study [21] that used VIA/VILI with the naked eye also found. The high sensitivity of VIA and the differences with the literature compared to the VILI or VIA/VILI examinations may have several explanations. First, in one study [3], the reviewers examined the cervical lesion images for a longer period of time and had the opportunity to compare the original images with the VIA image consecutively in order to establish differences in clinical results. In another study [5], the observers of the cervical lesion images were expert colposcopists, compared with novice physicians, who performed the evaluation on-site and off-site. Another possible explanation is that smartphone cameras (with high pixels) can focus on suspicious cervical lesions and detect cervical lesions with acetic acid whitening more easily than color changes produced by permeation with iodine in the VILI or VIA/VILI examinations. Moreover, in addition to post-VIA digital images, post-VILI digital images and native images can be archived and re-reviewed at any time. This evidence can contribute to improving diagnostic accuracy at the time of interpretation. However, this is not possible with the naked-eye inspection method, as, in clinical practice, once the VIA, VILI, or VIA/VILI examination has been performed, cervical lesions cannot be reinterpreted. Likewise, once Lugol’s iodine is applied, the cervix appears brown or black, and the native and acetic acid appearance can no longer be seen. It is also likely that the discrepancies between the studies are due to the VIA interpretation, which may vary between observers in terms of the different statistical methods used, including the sample size, the inclusion criteria of the studies, or the context and the level of care administered during the conduction of the studies. Finally, the smartphone-captured images were not necessarily simultaneously reviewed along with the VIA or VILI examination. Despite these explanations, unlike the studies that use naked-eye inspection, the use of a smartphone for the detection of cervical lesions following a VIA examination shows more favorable results, and the images can be reviewed at any time and by different specialists (experts or novices).

In our systematic review, the VIA/VILI combination with the use of a smartphone was less specific than that reported in a study [21], wherein the specificity was 91% with the VIA/VILI examination with the naked eye and ranged between 62.4% and 85.7%. The specificity of 85.7% for VIA/VILI examination in one study is probably due to the fact that the on-site physician and the three off-site physicians were experts [4]. On the contrary, another study reported low specificity [5], probably due to the fact that different physicians participated in the on-site and off-site trials while performing colposcopies and obtaining samples for histological study. It should also be noted that VIA using a smartphone was less specific (range, 24–50.4%) than that reported in a study wherein the specificity for VIA with the naked eye was 85% (95% CI, 81–89%) [23]. Similar to this, we observed low specificity for VILI (54.6%) with a smartphone compared with the high specificity of VILI with the naked eye reported in the literature (85–91.2%) [21,23]. These differences and the low specificity reported in the literature are attributable to various factors. First, the results with the smartphone VIA examination were more specific when they were performed by more experienced evaluators compared with the less experienced ones. Second, the high specificity of VILI with the naked eye compared with the low specificity of VILI with a smartphone might be related to the fact that VILI with the naked eye is easier to interpret than VIA or VIA/VILI [24], whose validity is based more on the experience and training of the health worker. On the other hand, some studies were conducted in a tertiary care hospital, where high-grade CIN lesion rates were higher when compared with screening at the primary level of care, which directly or indirectly influences the specificity [8].

Regarding the concordance indices of the group’s publications, Catarino [4] and Ricard-Gauthier [8] reported low levels of concordance between the interpretation of doctors on-site and off-site, which confirm that the reading of the VIA images with a smartphone may vary between different observers. On the other hand, Tanaka [5] reported a robust correlation between the histological diagnoses from images obtained by smartphone versus those obtained by colposcopy, a result that would be in line with the experience of the two colposcopists who participated in the study. However, differences were not always observed between the on-site vs. off-site results. In rural areas of Georgia, Ferris et al. [25] estimated the efficacy of telecolposcopy vs. traditional colposcopy, and the agreement of colposcopic impression with histology varied very little between groups. The agreement rate of on-site colposcopists was 59.7% (Κ: 0.31), 52.7% (Κ: 0.22) for local experts, and 55.7% (Κ: 0.27) for off-site experts who viewed the exam simultaneously. However, in Botswana, Quinley et al. evaluated the diagnostic agreement by visualizing remote cervical images (PIA) by an expert gynecologist off-site vs. the reading of the PIA images that were taken by a nurse on-site. The nurse’s positive reading was in agreement with that of the off-site expert in a significant number of cases [10]. A critical factor for obtaining suitable results when interpreting the images is the training of the health professional who obtains them. Finally, Schadel et al. evaluated the diagnostic reliability of a cervical examination with a digital colposcope vs. a conventional binocular colposcope based on two observers, and the agreement between both was 69% (Κ: 0.6). Both physicians had comparable diagnostic quality, 69% and 68%, for each case [26]. These findings may have different explanations. First, the poor concordance when observers use VIA may be due to different observers’ interpretations, the training that physicians receive, and the quality control of the images, as these are mandatory for achieving success when evaluating cervical lesions. Second, the moderate agreement between clinicians and nurses was probably due to ongoing training and permanent expert assistance as it is important for the clinicians to capture suitable quality VIA images. Moreover, the health professional or supervisor must be able to identify relevant cervical abnormalities in order to capture the images. Another explanation is that diagnostic concordance studies were conducted at different levels of care and regions, such as India, Madagascar, Japan, and Turkey; therefore, diagnostic concordance results may vary.

It should be noted that although HPV-PCR and PAP tests are considered to be more effective methods for the detection of cervical lesions, different studies show that costs and logistical barriers to their implementation present important challenges, especially in low- and middle-income countries [27]. In this sense, we recovered field trials that sought to overcome some of the barriers to the implementation of more complex tests (PCR-HPV and PAP). In our systematic review, we encountered studies that tested the use of smartphones with VIA, VILI, or VIA/VILI examinations and computer platforms for the detection of cervical lesions, as well as for their monitoring and control. These studies used these technologies for maximizing the early detection of ≥2 CIN lesions. In two studies [15,16], mobile colposcopes attached to smartphones were used. According to some authors, the smartphone image quality may not be as suitable as those obtained using colposcopy. However, smartphone images have several advantages, such as ease of use, low cost, storage of images (native, VIA, and VILI) for use at any time, fast delivery, allowing zooming in of the image, and no requirement for external light. Due to the advantages of smartphone images, compared to techniques that use the naked eye, the clinical decision could be based on the images captured with smartphones, especially in low-resource settings or first levels of care (secondary prevention strategy). This could be described because several studies were carried out in low-income countries, where there are scarce medical resources or nurses specialized in the detection of UCC, and limited technological resources for the provision of health services, unlike the two studies that were carried out in Japan [5] and the USA [16] where consultations for remote diagnosis are more common. Moreover, another study [15] showed that the use of a computer platform in an app can assist in the development of machine learning algorithms to improve the quality of care and support for clinical decisions. This algorithm is known as automated visual evaluation (AVE) and consists of a mobile colposcope built around a smartphone, an app, and a repository that stores digital images of the cervix to detect cancerous or precancerous lesions. It can even help when the PAP result is negative; a suspicious AVE image could provide valuable additional information for cervical lesion screening. This technology is being widely used in at least 17 countries.

Our study has several limitations. The first is that only five studies that evaluated sensitivity and specificity were included, and these studies were from Madagascar and Japan. Consequently, these results may be limited to Madagascar and Japan and may not be generalizable across other countries. Second, the heterogeneity of the procedures that were used could influence the variability of the sensitivity and specificity rates estimated in our study. Third, the studies did not assess the quality of the smartphone-captured images, which could lead to false-positive or -negative results that were not controlled. Fourth, in several studies, a visual evaluation was performed in HPV-positive women to identify possible cases with ≥2 CIN lesions. The final limitation is the small sample size included in the studies that only verified the usefulness of the smartphone in the diagnosis of cervical cancer. Despite these limitations, the QUADAS-2-based analysis of the quality of the studies showed a low risk of bias (20%) and a high applicability rate (100%).

## 5. Conclusions

Our findings suggest that the smartphone could be useful in the detection of uterine cervical lesions; however, its sensitivity and specificity are still limited. Despite this, we did ascertain some important findings. First, a VIA using a smartphone is more sensitive than a VILI or VIA/VILI combination for the detection of uterine cervical lesions. Second, a VIA using a smartphone is more sensitive than VIA, VILI, or VIA/VILI examinations with the naked eye; thus, it can improve diagnostic accuracy for the detection of ≥2 CIN lesions. Third, using a smartphone has several advantages, such as ease of use, image storage (native, after acetic acid application, and Lugol’s iodine) for use at any time, fast delivery for reviewers, and low cost. Fourth, due to the advantages of VIA with a smartphone, this method could be cost-effective in low-income settings where clinicians or colposcopes are not available, or it could be easily integrated into a cervical cancer screening program as a complement to a VIA with the naked eye. Finally, more research is needed to confirm and improve the usage of smartphones for UCC diagnosis.

## Figures and Tables

**Figure 1 cancers-13-06047-f001:**
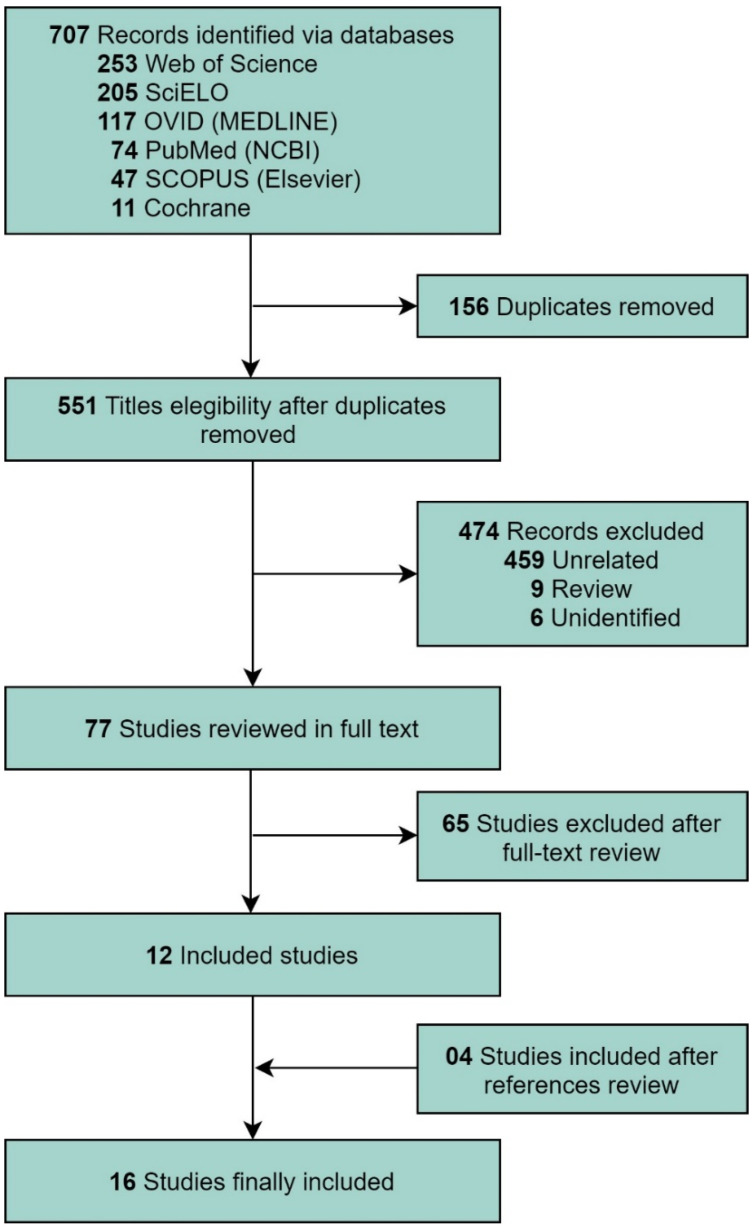
Flow diagram of the Preferred Reporting Items for a Systematic Review and Meta-analysis guidelines for the systematic literature search.

**Figure 2 cancers-13-06047-f002:**
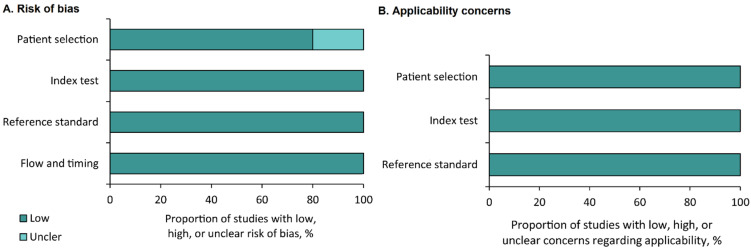
Summary of the results of the quality assessment using Quality Assessment of Diagnostic Accuracy Studies-2. Risk of bias (**A**) and applicability concerns (**B**).

**Figure 3 cancers-13-06047-f003:**
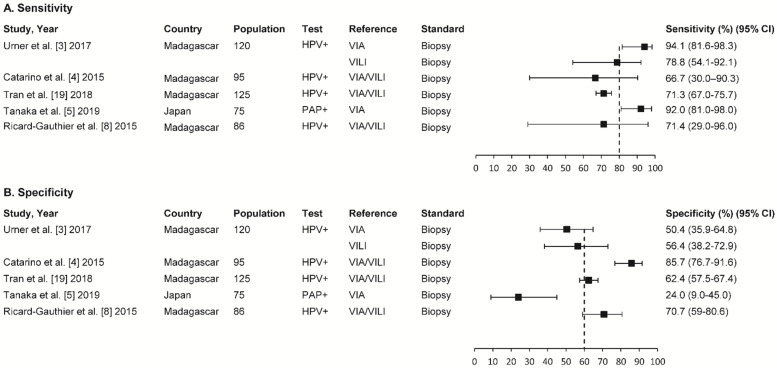
Summary of sensitivity (**A**) and specificity in five studies (**B**). VIA: visual inspection with acetic acid; VILI: visual inspection with Lugol’s iodine; CIN: cervical intraepithelial neoplasia; CI: confidence interval.

**Table 1 cancers-13-06047-t001:** Search strategy of papers in databases.

Database	Search Strategy
PubMed	((((uterine cervical cancer [Title/Abstract]) OR (cervical cancer screening [Title/Abstract])) AND (mobile application [Title/Abstract])) OR (tele-cytology [Title/Abstract])) OR (telediagnosis [Title/Abstract])
SCOPUS	TITLE-ABS-KEY (smartphones) OR TITLE-ABS-KEY (cellphones) OR TITLE-ABS-KEY (telecitology) OR TITLE-ABS-KEY (telehealth) OR TITLE-ABS-KEY (“Mobile technology”) AND TITLE-ABS-KEY (“uterine cancer”) OR TITLE-ABS-KEY (“cervical cancer”)
WoS	(cervical cancer OR uterine cervical cancer) AND (app OR mobile OR smartphone OR mobile technology OR tele-cytology OR telediagnosis)
OVID	(cervical cancer OR uterine cervical cancer OR cervical cancer screening) AND (mobile OR smartphone OR mobile technology OR tele-cytology OR telediagnosis)
SCIELO	#1 Expression: (uterine cervical cancer screening) AND (mobile technology) OR (telediagnosis) OR (tele-cytology) OR (smartphone) OR (mobile application) #2 Expression: (uterine cervical cancer screening) AND (mobile technology) OR (telediagnosis) OR (tele-cytology) OR (mobile application)
Cochrane	(uterine cervical cancer screening) AND (mobile technology) OR (telediagnosis) OR (tele-cytology) OR (smartphone) OR (mobile application)

**Table 2 cancers-13-06047-t002:** Concordance analysis results.

Study, Year	Country	Population	Standard Test	Reference Test	Concordance Index (95% CI)
Catarino et al. [4] 2015	Madagascar	95 HPV (+) women	VIA and VILI	Biopsy	Physicians on-site (17.7%) vs. physicians off-site (21.7%), agreement rate 76% (Κ: 0.28)
Ricard-Gauthier et al. [8] 2015	Madagascar	86 HPV (+) women	VIA and VILI	Biopsy	Physician on-site vs. off-site observers (Κ: 0.29)
Tanaka et al. [5] 2019	Japan	75 HPV (+) women	VIA	Biopsy	Histological diagnosis with a smartphone vs. colposcopy (Κ: 0.67)
Sharma et al. [9] 2018	India	180 women over 30 years of age	25 nurses (13.8%) VIA (+) vs. expert physicians 32/180 (17.8%) VIA (+)	N/A	Nurses vs. expert physicians 0.45 (CI: 0.26–0.63)
Gallay et al. [13] 2017	Madagascar	Images from HPV (+) women aged 30 to 65 years	Image quality with VIA in women that are HPV (+) acquired with a smartphone using the EXAM app, evaluated by three expert physicians	N/A	Individual opinion of physicians vs. the consensus of the three physicians 0.45 (CI: 0.23–0.58)
Bagga et al. [11] 2016	India	230 women between 30 and 65 years	ColpPhon vs. colposcope	Histological testing	Image quality had an agreement in 82% (184/225) and in the diagnosis had an agreement in 90% (208/230)
Quinley et al. [10] 2011	Botswana	95 HIV-positive women	Interpretation of PIA by on-site nurse vs. gynecologist off-site	Image reading by gynecologist in 64/95 women	Concordance (+) 0.82 Concordance (−) 0.89
Sahin et al. [14] 2018	Turkey	42 women	Microscopic cytopathological diagnoses vs. smartphone static image diagnoses	N/A	Concordance 85.5% and discordance 20.44%
Singh et al. [12] 2020	India	186 women with positive Pap tests and a Swedish score ≥ 5	Evaluated by doctor A using a gynocular and doctor B with standard colposcopy	Histological testing	Swedish score of doctors A and B (Κ: 0.795)

Abbreviations: N/A: Not applicable; HPV: human papillomavirus; VIA: visual inspection with acetic acid; VILI: visual inspection with Lugol’s iodine; HIV: human immunodeficiency virus; PIA: photographic inspection with acetic acid; Pap, Papanicolau; CI: confidence interval.

**Table 3 cancers-13-06047-t003:** Usefulness of a smartphone in detecting cervical lesions.

Study, Year	Country	Study Population	Study Type	Mobile Technology Used	Control Group	Result
Peterson et al. [15] 2017	Eastern Africa	824 women screened in field 1 and 234 in field 2	Transverse	The MobileODT (Mobile Colposcope) Enhanced Visual Assessment System used by nurses	N/A	Field 1. 12.6% of 824 women had precancerous lesions, and 0.7% had suspected cancer. Field 2. Of 234 women, 4.7% had precancerous lesions, and 3% had suspected cancer
Madiedo et al. [16] 2017	USA	59 women	Transverse	Enhanced visual mobile colposcope used by expert colposcopists	Cytology	Imaging with the mobile colposcope can be useful in detecting inaccurate PAP results
Yeates et al. [17] 2016	Tanzania	1072 sexually active women between the ages of 25 and 49 years	Transverse	VIA enhanced by smartphone cervicography	The control was the opinion of an external consultant who reviewed the cervigram images sent remotely	The agreement rate between students and expert reviewers was 96.8%
Yeates et al. [18] 2020	Tanzania	10,545 women aged 25 to 49 years evaluated using SEVIA at 24 health facilities in five regions of Tanzania	Transverse	An enhanced VIA platform for smartphones for the secure real-time exchange of cervical images for remote support supervision and data monitoring and evaluation	N/A	VIA (+) rates during the first 6 months increased compared with the rates over a 6-month period in the previous year (before the introduction of SEVIA) among HIV+ and HIV− participants as well as first-time participants. However, they did not compare VIA results against a histological diagnosis
Tanaka et al. [20] 2017	Japan	20 women with abnormal cervical cytology	Pilot study to evaluate the usefulness of a smartphone for the diagnosis of CIN or invasive cancer	iPhone 5S (Apple, Los Altos, CA, USA) was called Smartscopy	Standard colposcopy	85% of CIN 1 cases and 100% of CIN 2 cases could be diagnosed with a smartphone

Abbreviations: N/A: Not applicable; HPV: human papillomavirus; VIA: visual inspection with acetic acid; VILI: visual inspection with Lugol’s iodine.; HIV: human immunodeficiency virus; CIN: cervical intraepithelial neoplasia.

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
