# Peer review of "Use of Smartphones for the Detection of Uterine Cervical Cancer: A Systematic Review"

_cancers, 2021, doi:10.3390/cancers13236047_

Round 1
Reviewer 1 Report
Dear Authors,
below these are some suggestions for You:
Material and methods.
- Good for you, that three scholars were involved in independent screening of the titles, abstracts, and the full texts. It provides reliability for your research.
Results:
- Fig. 1. Check numbers, there are some flaws in calculations
- Line 149 and 167. Numbers of Tables in a text are mistaken
Discussion:
- Line 303. AVE - Assessment or Evaluation?
- Line 305. EVA? Do you mean Enhanced Visual Assessment or Automated Visual Evaluation?
Best regards and good luck
Author Response
We thank the Reviewer for your comments and constructive criticism, we believe that the quality of our manuscript has been significantly improved. We have revised our paper in a point-by-point manner. Modifications are in yellow text.
Material and methods.
Comment 1: Good for you, that three scholars were involved in independent screening of the titles, abstracts, and the full texts. It provides reliability for your research.
Response 1: Thank you for your comment.
Results:
Comment 2: Fig. 1. Check numbers, there are some flaws in calculations
Response 2: Thank you for your comment. We have corrected this error.
Comment 3: Line 149 and 167. Numbers of Tables in a text are mistaken
Response 3: Thank you for your comment. We have corrected these errors.
Discussion:
Comment 4: Line 303. AVE - Assessment or Evaluation?
Response 4: Thank you for your comment. We have corrected this error.
Comment 5: Line 305. EVA? Do you mean Enhanced Visual Assessment or Automated Visual Evaluation?
Response 5: Thank you for your comment. We have corrected this error “automated visual evaluation”.
Reviewer 2 Report
The authors did a great job to perform the systematic review to examine the efficacy of smartphones for the detection of uterine cervical cancer. This manuscript is well written and I have some suggestions to improve the manuscript.
Table 1
The authors may use MeSH keyword in the PubMed and Cochrane search. Please use MeSH keywords.
Indeed most of the PRISMA guidelines have been met, but the authors have not included the completed checklist to accompany the main text nor do the authors mention the preregistration of the systematic review in for instance PROSPERO.
Has an information specialist been involved in the design of the strategy? If so, mention his/her contribution in the methods and acknowledgements.
Although all search strategies are shown in full, the PubMed and CENTRAL search could be reproduced by this reviewer. The searches yielded about twice as many as the reported hits.
For instance, performing the PubMed search according to the strategy shown in the supplement yielded 206 records, not 73. Please check the search terms.
Overall, this manuscript is well written and well organized. Some modifications in the methodology are recommended before publication.
Author Response
We thank the Reviewer for your comments and constructive criticism, we believe that the quality of our manuscript has been significantly improved. We have revised our paper in a point-by-point manner. Modifications are in yellow text.
Comment 1: The authors did a great job to perform the systematic review to examine the efficacy of smartphones for the detection of uterine cervical cancer. This manuscript is well written and I have some suggestions to improve the manuscript.
Response 1: Thank you for your comment.
Comment 2: Table 1. The authors may use MeSH keyword in the PubMed and Cochrane search. Please use MeSH keywords.
Response 2: We used Title/Abstract instead MeSH terms, because it allowed to retrieve a greater number of published articles. In addition, the terms "tele-cytology", "telediagnosis" and "mobile application" are not registered as MeSH terms.
Comment 3: Indeed most of the PRISMA guidelines have been met, but the authors have not included the completed checklist to accompany the main text nor do the authors mention the preregistration of the systematic review in for instance PROSPERO.
Response 3: Thank you for your comment. We have included PRISMA checklist (see Appendix A. Table S1). We do not register the study Protocol in PROSPERO. Reviews that have started or completed data extraction are not eligible for inclusion in PROSPERO (It is PROSPERO policy). The protocol is available at the request of the authors (in Spanish) (Appendix A. Table S1. PRISMA checklist).
Comment 4: Has an information specialist been involved in the design of the strategy? If so, mention his/her contribution in the methods and acknowledgements.
Response 4: Thank you for your comment. We have included your suggestions "Design of the search strategy, M.C.R-S".
Comment 5: Although all search strategies are shown in full, the PubMed and CENTRAL search could be reproduced by this reviewer. The searches yielded about twice as many as the reported hits.
Response 5: Thank you for your comment. The PubMed database were searched for articles published between January 1, 2010, and September 30, 2020. PubMed results in this study are those retrieved on that date.
Comment 6: For instance, performing the PubMed search according to the strategy shown in the supplement yielded 206 records, not 73. Please check the search terms.
Response 6: Thank you for your comment. The PubMed database were searched for articles published between January 1, 2010, and September 30, 2020. PubMed results in this study are those retrieved on that date.
Comment 7: Overall, this manuscript is well written and well organized. Some modifications in the methodology are recommended before publication.
Response 7: Thank you for your comment.
Round 2
Reviewer 2 Report
The authors revised the manuscript well.